# "Antimicrobial utilization in a paediatric intensive care unit in India: A step towards strengthening antimicrobial stewardship practices"

Madhusudan Prasad Singh[1], Nitin Rewaram Gaikwad[1]*, Yogendra Narayanrao Keche[1], Atul Jindal[2], Suryaprakash Dhaneria[3], Meenalotchini Prakash Gurunthalingam[1]

1 Department of Pharmacology, All India Institute of Medical Sciences, Raipur, Chhattisgarh, India,
2 Department of Paediatrics, All India Institute of Medical Sciences, Raipur, Chhattisgarh, India,
3 Department of Pharmacology, Ruxmaniben Deepchand Gardi Medical College, Ujjain, Madhya Pradesh, India

* nitingaikwad2707@aiimsraipur.edu.in

**Data Availability Statement:** All relevant data are within the manuscript and its Supporting

## Abstract

Antimicrobials are frequently used in critically ill children admitted to the Paediatric Intensive Care Unit (PICU). The antimicrobial use data from Indian PICUs is limited using standard metrics such as Days of therapy (DOT). This study aimed to determine the baseline trend of antimicrobial use in PICU of a tertiary care teaching hospital of Raipur district of Chhattisgarh, India using standard metrics with the goal of developing facility-wide antibiotic policy and strengthening the antimicrobial stewardship activities. This active surveillance was conducted over a period of 18 months, from November 1, 2019, to March 21, 2021, in patients aged one month to 14 years who were admitted for $\geq$ 48 hours to the PICU at a tertiary care teaching hospital of Raipur District. Data on patient characteristics, antimicrobial indications, antimicrobial prescription information, and clinical outcomes were collected using pre-designed data abstraction forms. The descriptive statistic was used to represent the results. The antimicrobial consumption was analyzed according to the WHO AWaRe Class (Access, Watch, and Reserve groups) of antibiotics. The antimicrobial consumption was expressed as DOT/1000 patient-days (PD). A total of 216 patients were surveyed during the study period. The average number of antimicrobials prescribed per hospitalisation was 2.60 (range: 1–12), with 97.22% administered via parenteral route. Overall, DOT/1000-PD was 1318. The consumption of Watch Group antimicrobials was highest with 949 DOT/1000-PD, followed by Access (215) and Reserve Group (154), respectively. Ceftriaxone (208 DOT/1000 PD) was the most commonly prescribed antimicrobial agent, followed by Vancomycin (201), Meropenem (175), Piperacillin-Tazobactam (122) and Colistin (91). The patients who were escalated (28.24%) from empirical antimicrobial therapy had longer median PICU stay (8 days) compared those who were de-escalated (23.6%). Targeted therapy was given in 10.2% patients. The overall mortality rate was 14.35% and was higher (29.3%) in patients in whom empirical therapy was escalated compared to those who were de-escalated or continued. The study established a benchmark for antimicrobials use in the PICU and highlighted

Information files. It include raw dataset in supporting information file.

**Funding:** The author(s) received no specific funding for this work.

**Competing interests:** The authors have declared that no competing interests exist.

priority areas for antimicrobial stewardship intervention to enhance de-escalation rates, enhance targeted therapy, and reduce the overuse of antimicrobials especially belonging to the reserve group.

## Introduction

According to World Health Organization (WHO), inappropriate antimicrobial use contributes to antimicrobial-resistance (AMR) worldwide. In South-East Asia, antimicrobial misuse is common and contributes to antimicrobial resistance [1]. The inappropriate use of broad-spectrum antimicrobials can lead to infections due to multi-drug-resistant gram-negative bacilli and invasive candidiasis [2, 3], Multi-drug-resistant organism (MDRO) infections increase healthcare costs, length of stay, Intensive care unit (ICU) admission, morbidity, and mortality [2, 4–6]. A WHO-funded community-based survey in India found that up to 53% of public sector primary care patients and 70% of private sector patients receive antimicrobials for upper respiratory tract infection and acute diarrhoea in children and adults, indicating overuse and inappropriate use [7].

Paediatric patients belong to special population group, therefore irrational and inappropriate antimicrobial use that causes AMR is a major concern. Primary infections that require hospitalisation or secondary infections, including hospital-acquired infections (HAIs), are the most common issues in a Paediatric Intensive Care Unit (PICU) [1]. Patients with severe sepsis, septic shock, and HAI need complete and appropriate antimicrobial therapy [8–10].

Antimicrobials are widely used in PICUs (67%–97%) due to empirical antimicrobial therapy [11]. Empirical antimicrobial therapy often begins with local susceptibility. However, signs, symptoms, and severity of the infection must be assessed and antimicrobial therapy narrowed for common ICU infections [12]. Inadequate documentation and/or negative culture-sensitivity reports, a deteriorating patient's clinical status, or a failure to communicate between clinicians or residents on clinical rounds often lead clinicians to escalate and change antimicrobials during empirical antimicrobial therapy [13].

Due to the massive AMR problem, focused and coordinated efforts to improve rational antimicrobial prescribing and active AMR surveillance through the Antimicrobial Stewardship Programme (AMSP) are needed. AMSP that has been successfully implemented in adults can also be implemented for paediatric populations. Studies by J R Paño-Pardo *et al.* [14], J Y Ting *et al.* [15], and Jef Willems *et al.* [16] have shown successful implementation of AMSP in paediatric settings. AMSP implementation requires baseline antimicrobial use and resistance patterns from our patient care areas. Antimicrobial use in children, especially in PICUs, has been poorly documented up until now. Upon reviewing the relevant literature, no Indian study has evaluated the use of antimicrobials in PICUs using standard metrics like Days of Therapy. Both antimicrobial use and clinical outcomes in PICU patients can provide insight. Thus, we designed this surveillance study to assess antimicrobial use and clinical outcomes in PICU-admitted patients receiving antimicrobial therapy.

## Materials and methods

The active surveillance was conducted over 18 months (from November 1, 2019 to April 30, 2021) in Paediatric Intensive Care Unit (PICU) of All India Institute of Medical Sciences (AIIMS), Raipur, Chhattisgarh, India. Ethics Committee approval and the consent waiver was obtained before initiating the surveillance (AIIMSRPR/IEC/2019/331) from Institute Ethics

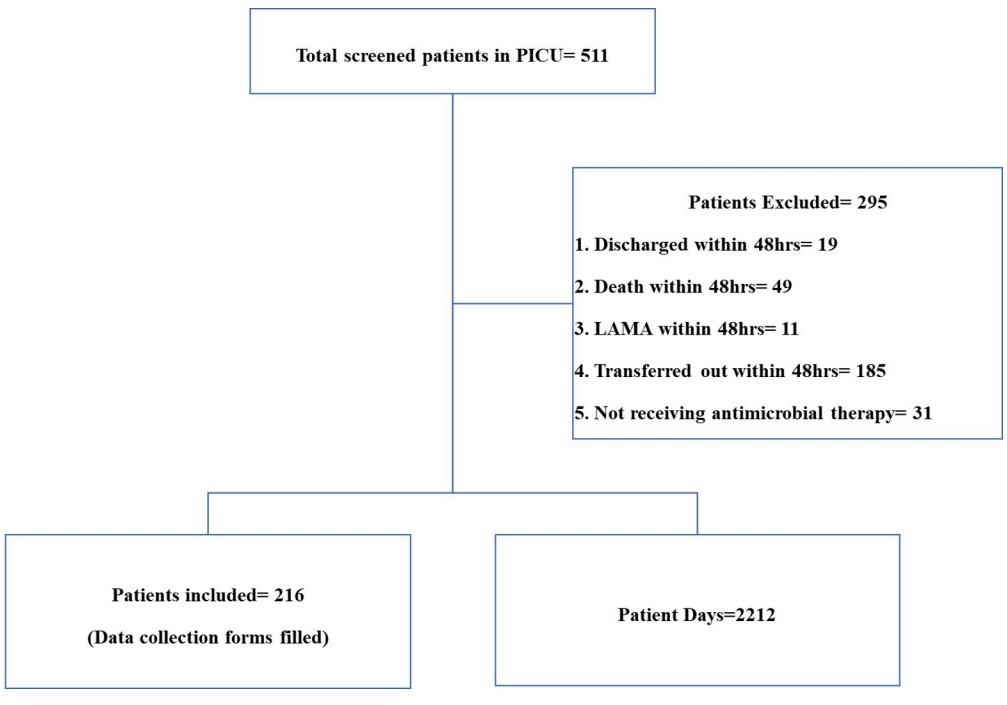

**Fig 1. Flow of the study participants.**

Committee. The patient identity was not revealed at any stage during and after the study. The privacy and confidentiality have been maintained.

All the admitted patients aged one month to fourteen years in PICU, receiving at least one or more antibacterial agents with or without antifungal agents and whose PICU stay was ≥ 48 hours were included. Patients who received only antitubercular, antifungal, or antiviral agents, topical antimicrobial agents, or who were discharged, left against medical advice (LAMA), or who died within 48 hours of admission were excluded. Active surveillance was conducted in the PICU every day between 11:00 AM and 1:00 PM. The data were collected using a pre-designed structured data abstraction form from the patients' health records, nursing charts and microbiology laboratory reports. The information collected comprise of antimicrobial agents given by system route (oral or parenteral), microbiological samples sent and their culture sensitivity reports, laboratory investigations, and patient-days. The patient-days were counted every day at 11 AM during the surveillance period and it included patients admitted to the PICU on or before 11:00 AM on the day of data collection (Fig 1).

## Data analysis

The data was analyzed for various parameters such as antimicrobial use expressed as DOT/1000 patient-days, WHO core prescribing indicators, number (%) of antimicrobials used empirically, number (%) of patients in who antimicrobial therapy de-escalated, escalated, or continued, number (%) of patients on targeted antimicrobials, number (%) of patients who received 'Access', Watch' or 'Reserve' antimicrobials and clinical outcomes like length of stay and mortality. To determine antimicrobial prescription compliance with the Essential Medicine List (EML), the list in use during the data collection period (National EML 2015 and WHO EML 2019) was applied. The operational definitions of above-mentioned terminologies can be found in the. (S1 File)

## Statistical analysis

Microsoft Excel 2013 were used to interpret the data for descriptive statistics such as median and interquartile range. Discrete data was expressed as counts or percentages. The antimicrobial consumption was expressed as DOT/1000 patient-days.

## Results

The study included 216 of 511 screened patients who met inclusion and exclusion criteria. At the end of the surveillance period, the total number of patient -days was 2212. (Fig 1) The patients had a median age of four years (IQR: 0.62–9.00). Table 1 depicts the demographic and clinical characteristics of the patients.

The average number of antimicrobial agents prescribed per hospitalisation was 2.60 (range: 1–12), with 97.22% given parenterally. The average duration of prescribed antimicrobial treatment was 5.20 ±1.30 days. During study period, 86.2% of antimicrobials were prescribed by generic names. The percentage of prescribed antimicrobials compliant with National EML, India (NLEM 2015) and WHO EML 2019 (21st Edition) were 69.1% and 80.6%, respectively. The patients received 94.88% of the prescribed antimicrobial dosages. Patients who received at least one antimicrobial agent accounted for 25% of the total, while 46% patients received three or more antimicrobials. The overall DOT /1000 patient-days were 1318. (Table 2) Ceftriaxone (208 DOT/1000 patient-days) was the most commonly prescribed antimicrobial agent, followed by vancomycin (201), meropenem (175), piperacillin-tazobactam (122) and amikacin

**Table 1. Demographic details and clinical characteristics.**

| Clinical Characteristics | Total (n = 216) |
|---|---|
| Age in years, Median (IQR) | 4.00 (0.62–9.00) |
| Gender, n (%) | |
| Male | 118 (54.63%) |
| Female | 98 (45.37%) |
| Management, n (%) | |
| Medical | 178 (82.4%) |
| Surgical | 38 (17.6%) |
| Mechanical Ventilation, n (%) | 47 (21.8%) |
| Average number of ventilator days | 7.68 days |
| Medical device in place, n (%) | 56 (25.9%) |
| Central line | 52 (24.07%) |
| Urinary Catheter | 48 (22.22%) |
| Febrile on admission, n (%) | 160 (74%) |
| Total Patient Days of PICU | 2212 |
| Reason for ICU admission (System-wise), n (%) | |
| Gastrointestinal | 10 (4.62%) |
| Liver and Biliary | 12 (5.55%) |
| Musculoskeletal | 09 (4.16%) |
| Hereditary/Congenital disorders | 18 (8.33%) |
| Respiratory | 60 (27.8%) |
| Renal/Urinary | 28 (12.9%) |
| CNS | 58 (26.8%) |
| Cardiovascular system | 26 (12.0%) |
| Oncology | 27 (12.5%) |
| General | 10 (4.62%) |

**Table 2. Antimicrobial usage data (in terms of DOT/1000 patient days and according to WHO AWaRe classification.**

**1. Antimicrobial usage data- DOT / 1000 patient days**

| Antimicrobial group | Antimicrobials | ATC Code | WHO AWaRe Classification | No. of patients (%) | Days of therapy (DOT) | DOT/1000 Patient days (% Consumption$) | Average antimicrobial treatment days |
|---|---|---|---|---|---|---|---|
| **All Antimicrobials** | | | | 216 (100%) | 2911 | 1318 | 5.20 |
| **Beta Lactams ± BLI** | | | | 104 (48.1%) | 469 | 212 (16%) | 4.5 |
| | Ampicillin | J01CA01 | Access group | 13 (6%) | 50 | 23 (1.7%) | 3.8 |
| | Ampicillin + Cloxacillin | J01CF02 | Access group | 14 (6.5%) | 67 | 30 (2.3%) | 4.8 |
| | Amoxicillin | J01CA01 | Access group | 1 (0.46%) | 3 | 1 (0.08%) | 3 |
| | Amoxicillin + Clavulanic Acid | J01CR02 | Access group | 19 (8.8%) | 64 | 29 (2.2%) | 3.4 |
| | Cloxacillin | J01CF02 | Access group | 3 (1.4%) | 15 | 7 (0.5%) | 5 |
| | Piperacillin + Tazobactam | J01CR05 | Watch group | 54 (25%) | 270 | 122 (9.3%) | 5 |
| **3rd Gen. Cephalosporins** | | | | 132 (61.11%) | 589 | 267 (20.3%) | 4.5 |
| | Ceftriaxone | J01DD01 | Watch group | 96 (44.4%) | 459 | 208 (15.8%) | 4.8 |
| | Cefotaxime | J01DD01 | Watch group | 26 (12%) | 100 | 45 (3.4%) | 3.8 |
| | Ceftazidime | J01DD02 | Watch group | 6 (2.6%) | 13 | 6 (0.45%) | 2.2 |
| | Cefpodoxime | J01DD13 | Watch group | 2 (0.9%) | 9 | 4 (0.3%) | 4.5 |
| | Cefixime | J01DD08 | Watch group | 1 (0.4%) | 2 | 1 (0.08%) | 2 |
| | Cefoperazone | J01DD12 | Watch group | 1 (0.4%) | 6 | 3 (0.2%) | 6 |
| Carbapenems | Meropenem | J01DH02 | Watch group | 59 (27.3%) | 386 | 175 (13.3%) | 6.5 |
| Monobactam | Aztreonam | J01DF01 | Reserve group | 2 (0.9%) | 9 | 4 (0.3%) | 4.5 |
| Polymyxins | Colistin | J01XB01 | Reserve group | 27 (12.5%) | 201 | 91 (6.9%) | 7.4 |
| **Glycopeptides** | | | | 81 (37.5%) | 515 | 233 (17.7%) | 6.4 |
| | Vancomycin | J01XA01 | Watch group | 72 (33.3%) | 444 | 201 (15.3%) | 6.2 |
| | Teicoplanin | J01XA02 | Watch group | 9 (4.16%) | 71 | 32 (2.4%) | 7.9 |
| Oxazolidinones | Linezolid | J01XX08 | Reserve group | 9 (4.16%) | 74 | 33 (2.5%) | 7.4 |
| Glycylcycline | Tigecycline | J01AA12 | Reserve group | 5 (2.3%) | 49 | 22 (1.7%) | 9.8 |
| **Fluoroquinolones** | | | | 3 (1.39%) | 7 | 3 (0.2%) | 2.3 |
| | Levofloxacin | J01MA12 | Watch group | 2 (0.9%) | 6 | 3 (0.2%) | 3 |
| | Ciprofloxacin | J01MA02 | Watch group | 1 (0.46%) | 1 | 0.45 (0.03%) | 1 |
| Phosphonic acid derivative | Fosfomycin (IV) | J01XX01 | Reserve group | 1 (0.46%) | 9 | 4 (0.3%) | 9 |
| **Aminoglycosides** | | | | 76 (35.2%) | 264 | 119 (9%) | 3.5 |
| | Amikacin | J01GB06 | Access group | 65 (30%) | 223 | 101 (7.7%) | 3.4 |
| | Gentamicin | J01GB03 | Access group | 11 (5.09%) | 41 | 19 (1.4%) | 3.7 |
| **Macrolides** | Azithromycin | J01FA10 | Watch group | 24 (11.1%) | 126 | 57 (4.3%) | 5.2 |
| **Tetracyclines** | Doxycycline | J01AA02 | Access group | 3 (1.4%) | 12 | 5 (0.4%) | 4 |
| **Rifamycin** | | | | 5 (2.3%) | 32 | 15 (1.1%) | 6.4 |
| | Rifampicin | J01AM02 | Watch group | 2 (0.9%) | 8 | 4 (0.3%) | 4 |
| | Rifaximin | A07AA11 | Watch group | 3 (1.4%) | 24 | 11 (0.8%) | 8 |
| **Lincosamide** | Clindamycin | J01FF01 | Access group | 9 (4.2%) | 37 | 17 (1.3%) | 4.1 |
| **Sulfonamides** | Sulfamethoxazole and trimethoprim | J01EE01 | Access group | 14 (6.48%) | 88 | 40 (3%) | 6.3 |
| **Nitroimidazole** | Metronidazole | J01XD01 | Access group | 10 (4.6%) | 34 | 15 (1.1%) | 3.4 |
| **Nitro-methylene-diamino imidazolidinediones** | Nitrofurantoin | J01XE51 | Access group | 1 (0.46%) | 10 | 5 (0.4%) | 10 |

**2. Antimicrobial usage as per WHO AWaRe Classification**

*(Continued)*

**Table 2.** (Continued)

| AWaRe Class | Days of Therapy/1000 Patient days (% Consumption$^\$$) | Average antimicrobial treatment days | Percentage of patients receiving antimicrobials* |
|---|---|---|---|
| Access group | 215 (16.32%) | 3.9 | 115 (53.2%) |
| Watch group | 949 (72.00%) | 4.9 | 198 (91.7%) |
| Reserve group | 154 (11.68%) | 7.6 | 29 (13.4%) |
| Total | 1318 | - | - |

**Note:** $^\$$Numerator: DOT/1000PD, $^\$$Denominator: Total DOT/1000PD (= 1318), *Numerator: Number of patients receiving at least one antimicrobial agent from Access/Watch/Reserve group, *Denominator: Total number of patients receiving antimicrobials (= 216)

(101) (Table 2) In 91.7% patients, at least one antimicrobial agent from 'Watch' group was prescribed. (Table 2) Colistin (91 DOT/1000 patient-days), linezolid (33), and tigecycline (22) were the most frequently prescribed antimicrobials in the reserve group. (Table 2) The DOT/LOT ratio was 1.92, with a total DOT of 2911 days and LOT of 1520 days. The average antimicrobial treatment days for 'Reserve' group antimicrobials were higher (7.6 days) than for 'Watch' (4.9 days) and 'Access' (3.9 days) antimicrobials. (Table 2) Redundant double gram-negative and double anaerobic coverage antimicrobial coverage was given in 27.3% and 20.80% of patients, respectively (Fig 2).

S3 Fig depicts empirical antimicrobial use. Ceftriaxone was empirically prescribed to 98.90% of patients, followed by vancomycin (82.20%) and amikacin (77.60%). The culture sensitivity report determined targeted antimicrobial therapy for 47.8% of culture positive isolates. (S1 Table) Empirical antimicrobial therapy was de-escalated in 23.6% of patients, 17.6% of whom had microbiological culture-sensitivity reports. Based on culture-sensitivity, 11.1% of patients had empirical antimicrobial therapy de-escalated, escalated, or continued. (Table 3) The median length of PICU stay for the patients who were de-escalated, escalated, and continued antimicrobial therapy was 7 days (IQR: 5–12.25 days), 8 days (IQR: 5–12 days), and 4 days (IQR: 3–6.75 days), respectively (Fig 3).

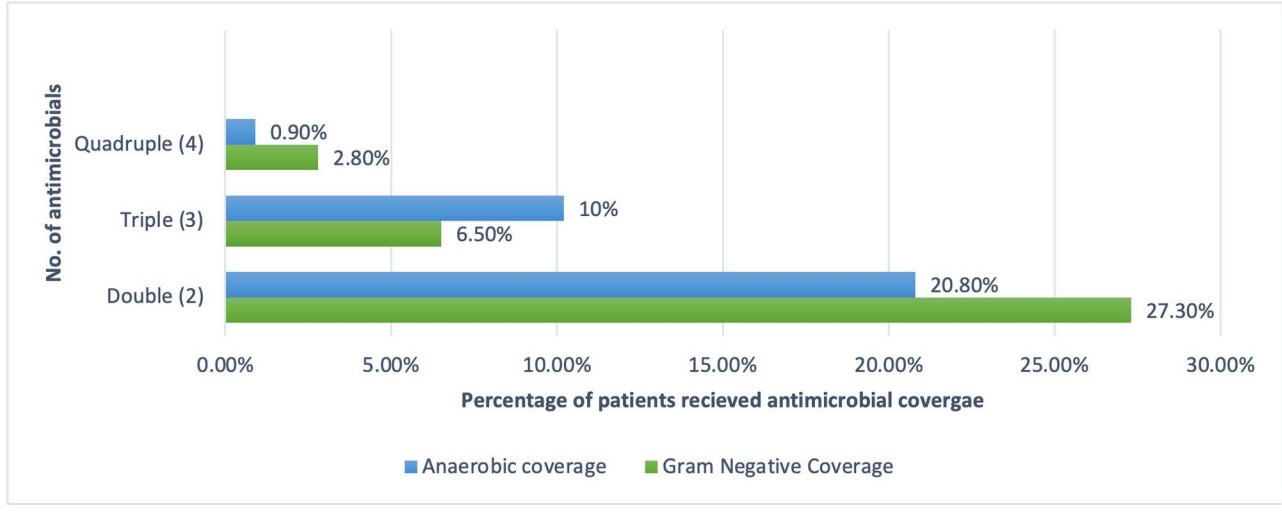

**Fig 2. Data showing percentage of patients who received multiple drug coverage for Gram negative and anaerobic infections.** Note: *Denominator: Total number of patients under surveillance (= 216).

**Table 3. De-escalation, escalation and continuation of antimicrobial therapy.**

| S.N. | | Parameter | Value |
|---|---|---|---|
| 1 | A | Percentage of patients in whom antimicrobial therapy was de-escalated, n (%) * | 51 (23.6%) |
| | B | Withdrawal of ≥1 antimicrobial agents | 51 (23.6%) |
| | C | Withdrawal of at least one antimicrobial agent + Addition of narrow spectrum antimicrobial agent/s | 13 (6.01%) |
| | D | Stopping empirical therapy and switching to narrow spectrum | 9 (4.16%) |
| | E | Percentage of patients in whom antimicrobial therapy was de-escalated based on culture sensitivity report, n (%) $ | 9 (17.6%) |
| 2 | A | Percentage of patients in whom antimicrobial therapy was escalated, n (%) * | 61 (28.24%) |
| | B | Addition of ≥1 antimicrobial agent/s to empirical therapy | 61 (28.24%) |
| | C | Switching from narrow spectrum to broad spectrum | 39 (18.05%) |
| | D | Withdrawal of ≥1 antimicrobial agent/s from empirical therapy BUT Addition of ≥1 broad spectrum antimicrobial agent/s | 31 (14.35%) |
| | E | Percentage of patients in whom antimicrobial therapy was escalated based on culture sensitivity report, n (%) $ | 6 (9.8%) |
| 3 | A | Percentage of patients in whom therapy was continued, n (%) * | 104 (48.15%) |
| | B | Percentage of patients in whom therapy was continued based on culture sensitivity report, n (%) $ | 9 (8.7%) |
| 4 | | Total Percentage of patients in whom therapy was De-escalated/escalated/continued based on culture sensitivity report, n (%) # | 24 (11.1%) |

**Note:** 1 = De-escalation categories, 2 = Escalation categories, 3 = Continuation categories, 4 = De-escalation, escalation or continuation based on culture-sensitivity report, Denominator: * Total number of patients under surveillance (= 216), $Total number of patients in whom therapy was De-escalated/escalated or continued (De-escalated = 51, escalated = 61, Continued = 104), # Total number of patients under surveillance (= 216)

At least one sample was sent for microbiological culture-sensitivity testing for 74.5% of patients. However, this testing was only done before empirical antimicrobial therapy for 46.8% of patients. The culture-positivity yield was 16.7% and the targeted therapy was administered for 47.8% of culture-positive isolates. (Table 3) The incidence rate of hospital-acquired MRSA, MDRO infection and multi-drug-resistant Gram-negative bacteria (MDR-GNB) infection per 1000 patient-days was 0.9, 6.3 and 4.1, respectively. (Table 4) The most common isolated organism was *Escherichia coli*– 10 (21.7%) followed by *Acinetobacter*– 9 (19.6%) *and Coagulase-negative Staphylococcus (CoNS)*– 7 (15.2%). (S1 Table, S2 and S3 Figs)

Of 216 patients, 69% received antimicrobial therapy for ≤7 days, 24% for 8–14 days, and 7% for ≥15 days. The Median PICU stay of study population was six (06) days (IQR:4–9). (Table 4) The total mortality rate was 14.35% (Table 4), and it was higher (29.3%) in patients whose empirical therapy was escalated than in those whose empirical therapy was de-escalated or continued. (Fig 4) After 48 hours of antimicrobials, 55% of 76% febrile admissions became afebrile. Seven (3.24%) of 216 patients under surveillance were readmitted within 7 days of discharge/transfer. (Table 4)

## Discussion

Antimicrobials are essential in paediatric infectious disease management, given children's heightened vulnerability to infections and the potential severity of illness. Beta-lactams with or without beta-lactamase inhibitors, carbapenems, glycopeptides, aminoglycosides, and polymyxins were the most commonly used antimicrobials in our study.

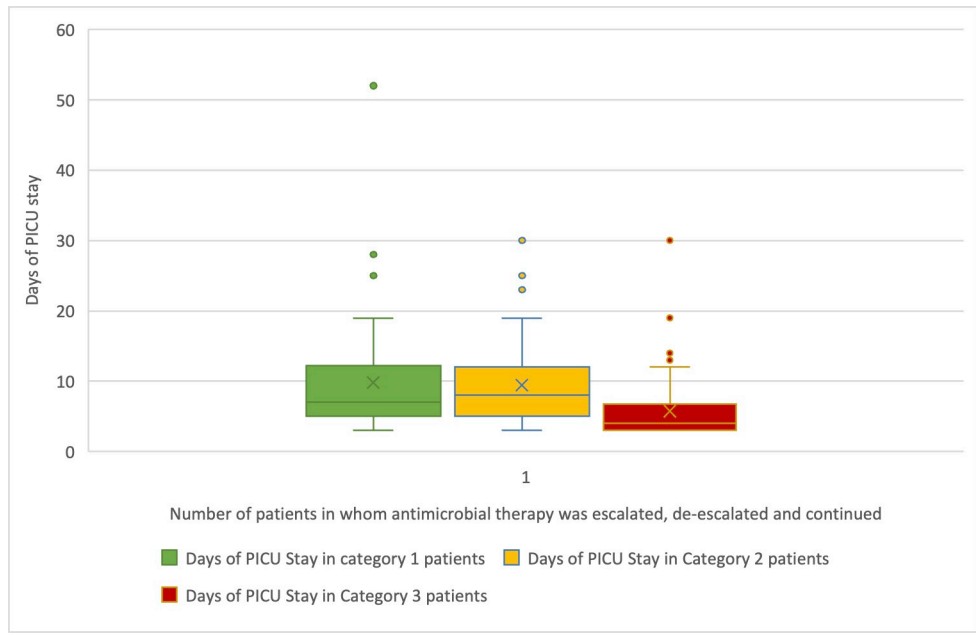

**Fig 3. Box and Whisker plot for PICU stay by number of patients in whom antimicrobial therapy was escalated, de-escalated and continue.** Note: Category 1 = No. of De-escalated Patients, Category 2 = No. of Escalated Patients, Category 3 = No. of Continuation therapy Patients.

In our study, we utilized metrics such as days of therapy (DOT) to assess antimicrobial usage within the paediatric intensive care unit (PICU) setting. DOT is advocated by the Centre for Disease Control (CDC) as the primary antimicrobial consumption metric due to its clinical relevance and suitability for facility benchmarking [8, 17]. Unlike defined daily doses (DDD), DOT accounts for changes in antimicrobial consumption due to age and weight modifications, making it more appropriate for paediatric populations [18, 19].

During the surveillance period, we calculated overall DOTs for different antimicrobial classes and individual agents based on the WHO AWaRe classification. Our analysis revealed a significant finding: a high level of antimicrobial usage in the PICU, with an observed consumption of 1318 DOT per 1000 patient days. This figure surpasses previous surveys conducted in other countries, indicating a notable discrepancy in antimicrobial consumption rates between different healthcare settings. A study conducted in three Paediatric and Neonatal Intensive Care Units in Saudi Arabia reported an antimicrobial consumption rate of 697 DOT per 1000 patient days, considerably lower than our findings [20]. Similarly, another study reported 1226 DOT per 1000 patient days in a PICU setting [21]. Comparisons with studies conducted in Western countries further underscore the magnitude of antimicrobial usage in our PICU. A study evaluating antimicrobial prescribing in PICUs across the United States found a median of 1043 DOT per 1000 patient days, indicating a lower but still substantial level of antimicrobial consumption compared to our findings [13]. Despite variations in methodology and geographical location, the consistently high antimicrobial consumption observed in our study raises concerns regarding antimicrobial stewardship practices in our facility.

'Watch' group antimicrobials like ceftriaxone, vancomycin, meropenem, piperacillin-tazobactam, and 'Reserve' group antimicrobial like colistin accounted for 60% of use. Balkhy *et al.* [20] reported the consumption trend of various antimicrobials in PICU in their 33 month surveillance study. The percentage share of consumption of these antimicrobials in our study, relative to total DOT per 1000 patient-days, is much less than the observations of Balkhy *et al.*

**Table 4. Clinical and microbiological indicators.**

| I. Antimicrobial treatment and PICU Stay | | |
|---|---|---|
| **Length of antimicrobial therapy** | ≤7 days | 148 (69%) |
| | 8–14 days | 52 (24%) |
| | ≥15 days | 16 (7%) |
| **Median PICU Stay (IQR) in days** | - | 6 (4–9) |
| **PICU Stay (Days)** | 2–10 days | 170 (79%) |
| | 11–20 days | 39 (18%) |
| | ≥21 days | 7 (3%) |

| II. Clinical outcome indicators | | |
|---|---|---|
| **S.N.** | **Parameter** | **Percentage** |
| **1.** | Mortality rate (n) [*] | 31 (14.35%) |
| **2.** | Percentage of patients who became afebrile within 48 hours of PICU admission[+] | 88 (55%) |
| **3.** | Readmission rate within 7 days of discharge/transfer out [*] | 7 (3.24%) |

| III. Microbiological Indicators | | |
|---|---|---|
| **1.** | Percentage of patients in whom sample was sent for culture sensitivity test [*] | 161 (74.5%) |
| **2.** | Culture positivity rate (Microbiological yield) [#] | 46 (16.9%) |
| **3.** | Number of samples in which multi drug resistant organisms (MDRO) was isolated, n (%) [$] | 30 (18.6%) |
| **4.** | Incidence Rate of Hospital- acquired MRSA infection per 1000 patient days[@] | 0.9 |
| **5.** | Incidence Rate of Total Hospital- acquired Multi drug resistant organism (MDRO) infection per 1000 patient days[@] | 6.3 |
| **6.** | Incidence Rate of Hospital- acquired Multi drug resistant gram-negative bacteria (MDR-GNB) infection per 1000 patient days[@] | 4.1 |

**Note:** Percentage rounded to nearest zero, [*] Denominator: Total number of patients under surveillance (= 216), [+] Denominator: Total number of patients who were febrile on admission (= 160), [#] Denominator: total number of samples sent (= 272), [$]Denominator: total number of patients in whom sample was sent for culture sensitivity test (= 161), [@]Denominator: Total patient days (= 2212)

[20] third-generation cephalosporins (20.25% vs. 38%); vancomycin (15.25% vs. 21.9%). However, percentage consumption share of carbapenem is comparable with this study (13.28% vs. 14%).

Although 'Reserve' group antimicrobials have lower consumption (91 DOT per 1000-patient days), the average antimicrobial treatment days of antimicrobials in the 'Reserve' group are considerably higher than those in the 'Watch' and 'Access' groups (7.6 days vs 4.9 days and 3.9 days). The findings emphasise the need for continuous monitoring and effective stewardship of 'Watch' and 'Reserve' group antimicrobials. In contrast to the observation by Panditrao *et al.* (2021) [22], our surveillance data indicated higher consumption of 'Watch' group antimicrobials (91.7% vs 80.56%), prominent being third-generation cephalosporins. Our data is limited to single centre and paediatric patients, unlike Panditrao *et al.* [22] observations from various treatment facilities and age groups; however, the widespread use of these antimicrobials raises their resistance potential, requiring periodic surveys. Most studies report paediatric antimicrobial usage ranging from 44% to 97%, indicating high rates [23–26].

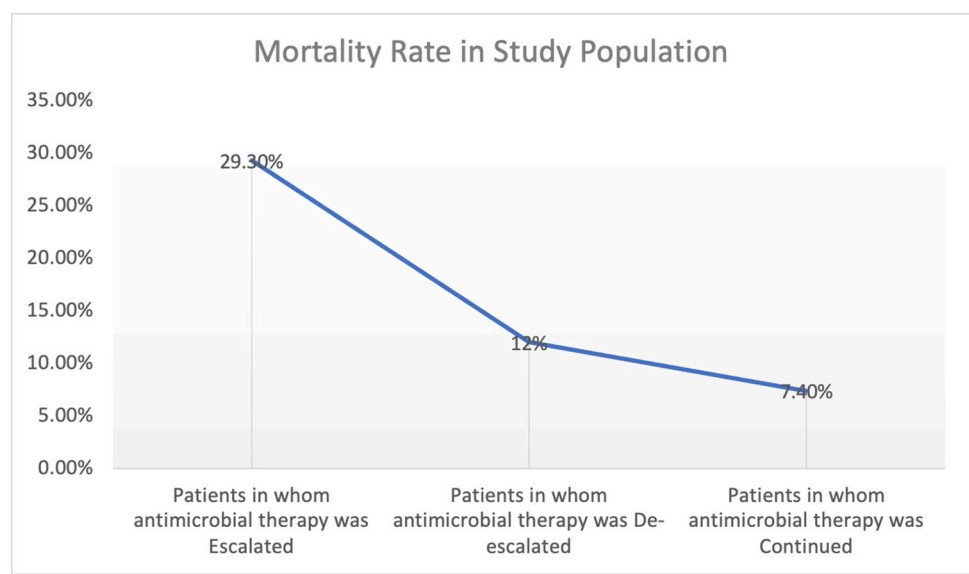

**Fig 4. Relationship between mortality rate and escalation, de- escalation and continuation of empirical therapy.**

In our study, the majority of the antimicrobials were prescribed empirically. Empirical use of ceftriaxone was highest with 98.90% among the 'Watch' group, followed by vancomycin (82.20%) and meropenem (67.80%). Not only this, empirical use of 'Reserve' group antimicrobials such as colistin (39.30%), linezolid (30%), tigecycline (20%) was also observed (S3 Fig). The use of 'reserve' antimicrobials is anticipated in critical care setting but high empirical use of these antimicrobials pose risk for development of AMR. Nonetheless, over-prescription is alarming and may be influenced by prior antimicrobial use, low culture positivity, critical illness, risk of MDRO infections, and non-improvement or clinical deterioration [27].

Antimicrobial consumption in a patient care setting reflects the prescribing practices of clinicians. In India, very limited data on antimicrobial consumption in paediatric patients is available in different treatment settings, including PICU. The prevalence of high antimicrobial use in paediatric patients in Southeast Asian countries is comparable to India and Western countries. In a study from Pakistan, Abbas *et al.* [11] (2016) found that 100% of PICU-admitted patients received at least one antimicrobial agent for prophylaxis, therapeutic, or empirical purposes. Similarly, Boone *et al.* [28] (2020) noted that antimicrobials were prescribed in 73% of infants admitted to hospitals in Bangladesh.

The non-availability of narrow-spectrum antimicrobial agents is an important deterrent for the rational use of antimicrobials. The use of amoxicillin-cloxacillin, amoxicillin-clavulanic acid, vancomycin, and linezolid for MSSA in place of cloxacillin is common because of its unavailability. However, cloxacillin is effective than vancomycin for MSSA bloodstream infections. Vancomycin should only be used if there is a life-threatening allergy [29]. In our study, the consumption of cloxacillin was meagre (7 DOT per 1000-patient days). One of two patients in whom MSSA was isolated had received the targeted therapy with cloxacillin.

In our study, we also discovered the unnecessary use of combination therapy, which is a cause of concern. DOT/LOT ratio is a useful indicator for the combination antimicrobial therapy. Overall, in our study, DOT/LOT ratio was 1.92 higher than the reported in other studies [30, 31].

Double Gram-negative and anaerobic coverage (DAC) was another finding in our surveillance. Song *et al.* [32] (2015) found that 26.8% of patients received unnecessary double

anaerobic coverage for over three days, compared to 20.8% in our study. In contrast, Ratta-naumpawan *et al.* [33] reported a cumulative incidence of 2.8% (22/781) and 1.9% (19/1002) DAC before and after AMSP changes.

Escalation of antimicrobial therapy correlated with higher mortality and extended PICU stays. Additionally, prescribing ≥ 3 antimicrobials and prolonged therapy were associated with longer PICU stays (S1 and S2 Figs). This study is the first to demonstrate from Indian PICU set-up, a correlation between prolonged hospital stays and increased mortality rates associated with the use of multiple antimicrobials. We observed a relationship between antimi-crobial escalation and higher mortality rates, as well as extended lengths of stay (LOS). Future research is warranted to conduct a detailed analysis of the various factors affecting mortality and LOS. Compliance with generic names and Essential Medicine List was suboptimal.

In our study, the hospital-acquired MRSA infection incidence rate was 0.9 per 1000 patient days, whereas VRSA and VRE rate was zero during the surveillance period. Qadri *et al.* [34] (2019), in their study from a PICU in India, reported a relatively higher MRSA rate of 1.6 per 100 admissions. In a study conducted by Alrabiah K *et al.* [35] in Saudi Arabia, the researcher analyzed three years' data in patients less than or equal to 14 years of age. In this study, of the total MRSA infected patients, 22% accounted for hospital-acquired MRSA infection.

We observed that the incidence rate of total hospital-acquired multi-drug resistant organ-ism (MDRO) infection and multi-drug resistant Gram-negative bacteria (MDR-GNB) infec-tion was 6.3 and 4.1 per 1000 patient days, respectively. Abbas Q *et al.* [11] reported an MDRO rate of 7%, which is comparable to our finding, and they attributed a high mortality rate of 9% to high MDRO rate in their study. El-Nawawy A *et al.* [36] also reported a high inci-dence of MRDO with more prevalence of Gram-negative bacteria in PICU, especially *Klebsi-ella* (30.5%), *Acinetobacter baumanii* (22.22%), and *Pseudomonas* (16.67%). Globally, the problem of MDRO is increasing. The Gram-negative organisms of concern are *Acinetobacter*, *Klebsiella pneumoniae*, *Pseudomonas aeruginosa*. These MDROs were also observed in our study. Therefore, stringent AMSP strategies to combat this problem is the need of the hour.

In the present study, the resistance pattern observed for meropenem was concerning, with 100% resistance in *Klebsiella*, 75% resistance in *Pseudomonas*, and 50% resistance in *Acineto-bacter* isolates. Notably, 25% of *Klebsiella* isolates exhibited resistance to colistin, which is con-sidered a last-resort antibiotic. These findings align with the alarming trend reported by the ICMR AMR network in India [37], where carbapenem resistance in *Acinetobacter baumannii* reached 87.8% in 2022, significantly limiting treatment options. The network also highlighted the high prevalence of carbapenem resistance in *Klebsiella pneumoniae* (75%) and *Acinetobac-ter baumannii* (88%) causing bloodstream infections in intensive care units (ICUs). Further-more, the report emphasized the concerning levels of oxacillin resistance in *Staphylococcus aureus* (87%) and vancomycin resistance in *Enterococcus faecium* (42%) isolates from ICUs. These findings underscore the urgency of implementing stringent infection control measures, particularly in critical care settings, to combat the escalating threat of antimicrobial resistance [37]. In addition, resistance pattern for different organisms is in. (S4 and S5 Figs)

Regular review of hospital formulary and alignment with the National Essential Medicine List can ensure the availability of narrow-spectrum antimicrobials for targeted drug therapy. The low culture-positivity yield in a critical care setting can be attributed to several factors. These include the lack of sampling before initiating antimicrobial therapy, improper sample collection techniques, drawing cultures from lines used for administering antimicrobials, inad-equate blood volume for testing in pediatric patients, and issues with transportation, incuba-tion, handling, and culture systems [38–42]. The culture-positivity yield was 16.9% in our study. Being a referral center, the patients received from other hospital often had received high-end antimicrobials. This low rate of culture-positivity has resulted in a reduced rate of

targeted therapy and de-escalation. Urgent implementation of stewardship interventions, such as promoting a culture of sending sample for culture-sensitivity–"culture of culture" and applying pharmacological concepts in prescribing, is necessary to improve these rates. Educational programs and feedback sessions for healthcare professionals should also be implemented to support these interventions.

The primary strength of this study is its 18-month longitudinal study design which provided a comprehensive assessment of antimicrobial consumption and clinical outcomes in a PICU using standard surveillance methodology and WHO-recommended indicators. However, challenges were encountered during the study, such as a smaller sample size due to the conversion of the PICU to a COVID ICU and delays in microbiological reports. The study primarily relied on microbiological data from patient files during the surveillance period. The appropriateness of antimicrobials could only be determined in a few cases due to low culture-positivity yield. Other indicators were used to assess appropriateness, but patient-specific clinical characteristics were not evaluated. The study suggests that more efforts are needed to improve rational prescribing practices. The challenges faced during data analysis for this study included the lack of comparable data from Indian PICUs and a scarcity of high-quality studies using standard antimicrobial consumption metrics. Categorizing patients for de-escalation and escalation was also challenging, especially when both occurred in the same patient on different occasions, requiring an agreement to address this issue. The Indian Academy of Pediatrics and ICMR collaborated in 2014 to combat antimicrobial resistance in children [43]. The CDC's Antibiotic Use (AU) Option in National Healthcare Safety Network (NHSN) offers an automated model for monthly DOT data collection [44]. Aligning with ICMR's initiative, a national network in India can provide baseline data for stewardship evaluation.

## Conclusion

This surveillance study has yielded invaluable insights into antimicrobial usage patterns within a pediatric intensive care unit (PICU) at a tertiary care hospital in India. By establishing a comprehensive baseline dataset, our findings not only provide crucial information for benchmarking antimicrobial usage across hospitals in the state and India but also highlight key areas for antimicrobial stewardship interventions. Specifically, our study identifies opportunities for optimizing antimicrobial therapy, including reducing redundant anaerobic coverage, enhancing culture-positivity yield for targeted therapy, and curbing the overuse of reserve group antimicrobials. Furthermore, our observations underscore the potential impact of appropriate antimicrobial use on clinical outcomes, such as mortality rates and length of stay in the PICU. Moving forward, the insights gleaned from this study can inform targeted interventions aimed at improving antimicrobial prescribing practices and ultimately enhancing patient care and safety in pediatric critical care settings.

## Supporting information

**S1 Table. Isolated pathogens and targeted antimicrobial therapy.** Note: B = Blood, U = Urine, TBA = Tracheobronchial aspirate, P = Pus, S = Sputum, ST = Stool. * Denominator taken is total number of isolates (= 46).
(DOCX)

**S1 Fig. Box and Whisker plot for PICU stay by number of antimicrobial agents prescribed.** Note: Category 1 = Number of patients who were prescribed 1 antimicrobial agent, Category 2 = Number of patients who were prescribed 2 antimicrobial agents, Category 3 = Number of

patients who were prescribed ≥3 antimicrobial agents.
(DOCX)

**S2 Fig. Box and Whisker plot for PICU stay by antimicrobial treatment days.** Note: Category 1 = Antimicrobial treatment days ≤7 days, Category 2 = Antimicrobial treatment days 8–14 days, Category 3 = Antimicrobial treatment days ≥15 days.
(DOCX)

**S3 Fig. Usage of antimicrobial agents as empirical therapy.** Note: *Denominator for 'overall percentage': Total number of patients under surveillance (= 216) and for 'percentage of patients prescribed as empirical therapy': Total number of patients who received respective antimicrobial agent.
(DOCX)

**S4 Fig. Stacked bar chart showing Gram positive organisms isolated and their sensitivity pattern for the antimicrobials in percentage.** NOTE: R = RESISTANT, S = SENSITIVE, I = INTERMEDIATE, P = PENICILLIN, OX = OXACILLIN, AM = AMPICILLIN, CRO = CEFTRIAXONE, FEP = CEFEPIME, COT = COTRIMOXAZOLE, VA = VANCOMYCIN, E = ERYTHROMYCIN, DO = DOXYCYCLINE, GM = GENTAMICIN, CM = CLINDAMICIN, NX = NORFLOXACIN, FM = NITROFURANTOIN.
(DOCX)

**S5 Fig. Stacked bar chart showing Gram negative organisms isolated and their sensitivity pattern for the antimicrobials in percentage.** NOTE: R = RESISTANT, S = SENSITIVE, I = INTERMEDIATE, P = PENICILLIN, AM = AMPICILLIN, AMC = AMOXICILLIN+-CLAVULANATE, TZP = PIPERACILLIN+TAZOBACTAM, CRO = CEFTRIAXONE, CTX = CEFOTAXIME, CAZ = CEFTAZIDIME, CXM = CEFUROXIME, CZ = CEFAZOLIN, CFM = CEFEXIME, FEP = CEFEPIME, COT = COTRIMOXAZOLE, VA = VANCOMYCIN, E = ERYTHROMYCIN, IPM = IMIPENEM, MEM = MEROPENEM, DO = DOXYCYCLINE, AN = AMIKACIN, GM = GENTAMICIN, TM = TOBRAMYCIN, CL = COLISTIN, ATM = AZTREONAM, CM = CLINDAMICIN, LVX = LEVOFLOXACIN, NX = NORFLOXACIN, FM = NITROFURANTOIN.
(DOCX)

**S1 File. Operational definitions.**
(DOCX)

**S2 File. Datasheet 1 and 2.**
(DOCX)

## Author Contributions

**Conceptualization:** Madhusudan Prasad Singh, Nitin Rewaram Gaikwad.

**Data curation:** Madhusudan Prasad Singh, Nitin Rewaram Gaikwad, Atul Jindal, Meenalotchini Prakash Gurunthalingam.

**Formal analysis:** Madhusudan Prasad Singh, Nitin Rewaram Gaikwad, Yogendra Narayanrao Keche.

**Investigation:** Madhusudan Prasad Singh.

**Methodology:** Madhusudan Prasad Singh, Nitin Rewaram Gaikwad.

**Project administration:** Nitin Rewaram Gaikwad, Suryaprakash Dhaneria.

**Resources:** Atul Jindal.

**Supervision:** Nitin Rewaram Gaikwad, Yogendra Narayanrao Keche, Suryaprakash Dhaneria.

**Validation:** Madhusudan Prasad Singh, Suryaprakash Dhaneria.

**Writing – original draft:** Madhusudan Prasad Singh, Nitin Rewaram Gaikwad.

**Writing – review & editing:** Madhusudan Prasad Singh, Nitin Rewaram Gaikwad, Yogendra Narayanrao Keche.

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
