## [Decision Letter · Decision Letter 0]

25 Apr 2024

PONE-D-24-03393"Antimicrobial utilization in a paediatric intensive care unit in India: a step towards strengthening antimicrobial stewardship practices"PLOS ONE

Dear Dr. Gaikwad,

Thank you for submitting your manuscript to PLOS ONE. After careful consideration, we feel that it has merit but does not fully meet PLOS ONE’s publication criteria as it currently stands. Therefore, we invite you to submit a revised version of the manuscript that addresses the points raised during the review process.

We look forward to receiving your revised manuscript.

Kind regards,

Petra Czarniak, PhD

Academic Editor

PLOS ONE

Journal Requirements:

Reviewers' comments:

Reviewer's Responses to Questions

**Comments to the Author**

1. Is the manuscript technically sound, and do the data support the conclusions?

Reviewer #1: Partly

Reviewer #2: Partly

2. Has the statistical analysis been performed appropriately and rigorously? 

Reviewer #1: N/A

Reviewer #2: No

3. Have the authors made all data underlying the findings in their manuscript fully available?

Reviewer #1: Yes

Reviewer #2: Yes

4. Is the manuscript presented in an intelligible fashion and written in standard English?

Reviewer #1: Yes

Reviewer #2: Yes

5. Review Comments to the Author

**Reviewer #1:** The study provides valuable insights into antimicrobial use in a pediatric ICU setting and underscores the importance of antimicrobial stewardship.

However, I would like to make some criticisms.

o Excluding patients who were discharged, left against medical advice, or died within 48 hours might skew the understanding of antimicrobial use patterns and outcomes in critically ill patients.

o The study primarily used descriptive statistics for analysis. Incorporating inferential statistical methods could provide deeper insights into the significance of the findings and the relationship between antimicrobial use and clinical outcomes.

o The study notes a high rate of empirical use of 'Reserve' antimicrobials, which is alarming. However, it does not delve deeply into the reasons behind this trend (severe infections? serious patients?) or suggest specific strategies for reducing their use.

o The low culture-positivity yield is acknowledged but not sufficiently analyzed.

o While the paper mentions the potential for resistance due to high use of 'Watch' and 'Reserve' antimicrobials, it lacks a detailed discussion on the observed or potential trends in antimicrobial resistance.

o The paper identifies areas for stewardship interventions but does not provide detailed strategies or frameworks for implementing these interventions (e.g. protocols of use?)

o The conversion of the PICU to a COVID ICU and its impact on antimicrobial use patterns is mentioned but not explored in detail.

**Reviewer #2:** I agree that your study is very important to establishing a baseline of current antimicrobial use in PICUs and to contribute to the overall data on AM use in order to establish antimicrobial stewardship programs. What was lacking in the current submission was more of a thorough review of your data and why some of your results were significant compared to other similar studies (even in the adult population). What does your data mean basically? As a reader I don't know what the normal antimicrobial use in Indian hospitals is. What do you mean by watch and reserve categories?

A more thorough explanation of some terminology would have been helpful to understand your results: the above terms watch and reserve categories of antimicrobials, de-escalating of antimicrobials are examples. Don't assume the reader will know what these mean.

Some of the tables you included were very busy and hard to read. I would suggest organizing them differently or splitting them into multiple tables if possible. Doing more stats on the data. Were there significant differences in the results.

You made some interesting observations and perhaps some conclusions without fully addressing (or including) some things. You noted that longer stays and mortality increased with more antimicrobials? was that because the patients were sicker? you should at least address that in your conclusion and either try to explain it or perhaps say that observation needs to be studied more.

Another observation was the low culture yield and you suggest that could be a result of previous therapy before they were admitted. Is there a limitation to checking if they had a positive result early on in that previous admission to support this? Could the culture result actually be a true negative? could there be inappropriate treatment just based on not trusting culture results?

6. PLOS authors have the option to publish the peer review history of their article (what does this mean?). If published, this will include your full peer review and any attached files.

Reviewer #1: No

Reviewer #2: No

---

## [Author Response · Author response to Decision Letter 0]

11 Jun 2024

Dear Reviewers,

We thank you for the insightful comments and suggestions. We have carefully considered each of the reviewers' comments, made appropriate corrections in the manuscript and have addressed the comments accordingly in the rebuttal letter.

we believe that the revisions made in response to the reviewers' comments have significantly strengthened the quality and clarity of our manuscript. We are confident that the revised version addresses all concerns raised by the reviewers and is now suitable for publication in PLOS One.

Thank you.

Sincerely,

Prof (Dr.) Nitin Rewaram Gaikwad 

Professor and Head

Department of Pharmacology

All India Institute of Medical Sciences, Raipur, Chhattisgarh, India

Nitingaikwad2707@aiimsraipur.edu.in

---

## [Editor Report · Decision Letter 1]

3 Sep 2024

"Antimicrobial utilization in a paediatric intensive care unit in India: a step towards strengthening antimicrobial stewardship practices"

PONE-D-24-03393R1

Dear Dr. Gaikwad,

We’re pleased to inform you that your manuscript has been judged scientifically suitable for publication and will be formally accepted for publication once it meets all outstanding technical requirements.

Kind regards,

Petra Czarniak, PhD

Academic Editor

PLOS ONE
---

## [Editor Report · Acceptance letter]

10 Sep 2024

PONE-D-24-03393R1 

PLOS ONE

Dear Dr. Gaikwad, 

I'm pleased to inform you that your manuscript has been deemed suitable for publication in PLOS ONE. Congratulations! Your manuscript is now being handed over to our production team.

Kind regards, 

on behalf of

Dr. Petra Czarniak 

Academic Editor

PLOS ONE